

# Modelled deposition of nitrogen and sulfur in Europe estimated by 14 air quality model-systems: Evaluation, effects of changes in emissions and implications for habitat protection

Marta G. Vivanco[1,] Mark. R. Theobald[1], Héctor García-Gómez[1], Juan Luis Garrido[1], , Marje Prank[2,3], Wenche Aas[4], Mario Adani[5], Ummugulsum Alyuz[6], Camilla Andersson[7], Roberto Bellasio[8], Bertrand Bessagnet[9], Roberto Bianconi[8], Johannes Bieser[10], Jørgen Brandt[11], Gino Briganti[5], Andrea Cappelletti[5], Gabriele Curci[12], Jesper H. Christensen[11], Augustin Colette[9], Florian Couvidat[9], Kees Cuvelier[13], Massimo D'Isidoro[5], Johannes Flemming[14], Andrea Fraser[15], Camilla Geels[11], Kaj M. Hansen[11], Christian Hogrefe[16], Ulas Im[11], Oriol Jorba[17], Nutthida Kitwiroon[18], Astrid Manders[19], Mihaela Mircea[5], Noelia Otero[20], Maria-Teresa Pay[17], Luca Pozzoli[6,21], Efisio Solazzo[21], Svetlana Tsyro[22], Alper Unal[6], Peter Wind[22,23] and Stefano Galmarini[21]

[1]Environmental Department, CIEMAT, Madrid, 28040, Spain
[2]Finnish Meteorological Institute, Helsinki, FI00560, Finland
[3]Cornell University, Ithaca, NY, 14850, USA
[4]NILU-Norwegian Institute for Air Research, Kjeller, 2007, Norway
[5]ENEA, Italian National Agency for New Technologies, Energy and Sustainable Economic Development (ENEA), Via Martiri di Monte Sole 4, 40129 Bologna, Italy
[6] Eurasia Institute of Earth Sciences, Istanbul Technical University, Turke
[7]SMHI, Swedish Meteorological and Hydrological Institute Norrköping, Norrköping, Sweden
[8]Enviroware srl, Concorezzo, MB, Italy
[9]INERIS, Institut National de l'Environnement Industriel et des Risques, Parc Alata, 60550 Verneuil-en-Halatte, France
[10]Institute of Coastal Research, Chemistry Transport Modelling Group, Helmholtz-Zentrum Geesthacht, Germany
[11]Department of Environmental Science, Aarhus University, Roskilde, DK-4000, Denmark.
[12]Department of Physical and Chemical Sciences, University of L'Aquila, L'Aquila, Italy
[13] Ex European Commission, Joint Research Centre JRC Institute for Environment and Sustainability I-21020 Ispra (Va), Italy
[14]European Centre for Medium-Range Weather Forecasts, Reading, UK
[15]Ricardo Energy & Environment, Gemini Building, Fermi Avenue, Harwell, Oxon, OX11 0QR, UK
[16]Computational Exposure Division, National Exposure Research Laboratory, Office of Research and Development, United States Environmental Protection Agency, Research Triangle Park, NC,
[17] BSC, Barcelona Supercomputing Center, Centro Nacional de Supercomputaci_on, Nexus II Building, Jordi Girona, 29, 08034 Barcelona, Spain
[18] Environmental Research Group, Kings' College London, London, UK
[19]Netherlands Organization for Applied Scientific Research (TNO), Utrecht, The Netherlands
[20]IASS, Institute for Advanced Sustainability Studies, Potsdam, Germany
[21] European Commission, Joint Research Centre (JRC, Ispra (VA), Italy
[22]Climate Modelling and Air Pollution Division, Research and Development Department, Norwegian Meteorological Institute (MET Norway), P.O. Box 43, Blindern, N-0313 Oslo, Norway
[23]Faculty of Science and Technology, University of Tromsø, Tromsø, Norway

*Correspondence to*: Marta G. Vivanco (m.garcia@ciemat.es)





**Abstract.** The evaluation and intercomparison of air quality models is key to reducing model errors and uncertainty. The projects AQMEII3 and EURODELTA-Trends, in the framework of the Task Force on Hemispheric Transport of Air Pollutants and the Task Force on Measurements and Modelling, respectively, (both task forces under the UNECE Convention on the Long Range Transport of Air Pollution, LTRAP) have brought together various regional air quality models, to analyze their performance in terms of air concentrations and wet deposition, as well as to address other specific objectives.

This paper jointly examines the results from both project communities by inter-comparing and evaluating the deposition estimates of reduced and oxidized nitrogen (N) and sulfur (S) in Europe simulated by 14 air quality model-systems for the year 2010. An accurate estimate of deposition is key to an accurate simulation of atmospheric concentrations. In addition, deposition fluxes are increasingly being used to estimate ecological impacts. It is, therefore, important to know by how much model results differ, and how well they agree with observed values, at least when comparison with observations is possible, such as in the case of wet deposition.

This study reveals a large variability between the wet deposition estimates of the models, with some performing acceptably (according to previously defined criteria) and others underestimating wet deposition rates. For dry deposition, there are also considerable differences between the model estimates. An ensemble of the models with the best performance for N wet deposition was made and used to explore the implications of N deposition in conservation of protected European habitats. Exceedances of empirical critical loads were calculated for the most common habitats at a resolution of 100×100 m2 within the Natura 2000 network, and the habitats with the largest areas showing exceedances are determined.

Moreover, simulations with reduced emissions in selected source areas indicated a fairly linear relationship between reductions in emissions and changes in deposition rates of N and S. An approximately 20% reduction in N and S deposition in Europe is found when emissions at a global scale are reduced by the same amount. European emissions are by far the main contributor to deposition in Europe, whereas the reduction in deposition due to a decrease of emissions in North America is very small and confined to the western part of the domain. Reductions in European emissions led to substantial decreases in the protected habitat areas with critical load exceedances (halving the exceeded area for certain habitats), whereas no change was found, on average, when reducing North American emissions, in terms of average values per habitat.

## 1 Introduction

Improvements have been made in reducing ecosystem exposure to excess levels of acidification in past decades, largely as a result of declining $SO_2$ emissions. However, in addition to acidification, emissions of $NH_3$ and $NO_x$ have altered the global nitrogen cycle, resulting in excess inputs of nutrient nitrogen into terrestrial and aquatic ecosystems (Maas &. Grennfelt, 2016). This oversupply of nutrients can lead to eutrophication and subsequent loss of biodiversity. With the aim of ensuring the long-term survival of Europe's most valuable and threatened species and habitats, the Natura 2000 network of protected areas (EEA, 2017) was established in Europe under the 1992 Habitats Directive (EU, 1992). While it is



estimated that only 7% of the total EU-28 ecosystem area and 5% of the Natura 2000 area was at risk of
acidification in 2010 (EEA, 2015), it is estimated that the fraction exposed to air-pollution levels
exceeding eutrophication limits is 63% and 73%, respectively, in 2010 (EEA, 2015).

The Task Force on Hemispheric Transport of Air Pollution (HTAP) under the UNECE Convention on
Long Range Transport of Air Pollution program (CLRTAP) has organized several modeling exercises to
understand the role of hemispheric transport when estimating the impacts of remote sources on
background concentrations and deposition in different parts of the world (Galmarini et al. 2017). A
description of the HTAP program can be found at www.htap.org. While early exercises used global
models, the most recent research activity, HTAP2, foresees a combination of global and regional models,
in order to evaluate air pollution impacts at a higher spatial resolution. In this context, the project
AQMEII (Air Quality Model Evaluation International Initiative, Rao et al. 2009) in its third phase activity
(AQMEII 3) has brought together various air quality modelling teams from North America and Europe to
conduct a set of the simulations under the HTAP framework (Solazzo et al. 2017). At the same time, the
EURODELTA-Trends (EDT) project has also brought together several European modeling teams, to
provide information for the Task Force on Measurements and Modelling (also under the CLRTAP),
including the evaluation of models for specific campaigns (Bessagnet et al. 2016; Vivanco et al, 2016),
and, more recently, for 20-year trends of air quality and deposition (Colette et al. 2017). Since both
projects have a model evaluation component and there is a common simulation year (2010), it is possible
to evaluate the datasets jointly, enabling the comparison of a larger number of models (eight for
AQMEII3 plus seven for EDT).
The availability of 14-model simulations provides the possibility of obtaining a more robust ensemble
model estimate of deposition than that from a single model, as well as an estimate of deposition
uncertainty. This more robust estimate is particularly useful for assessing ecological impacts such as
critical load exceedance. Critical loads (CL) are limits for deposition of atmospheric pollutants, set by the
Working group on Effects of the CLRTAP for the protection of ecosystems (de Wit et al., 2015).
Exceedances of CL have been utilized during the last decades to assess impacts of atmospheric pollution
to natural and semi-natural European ecosystems. Moreover, applying empirical CL for the nutrient N is
recommended to assess "whether N deposition should be listed as a threat to future prospects" in the
framework of the Habitats Directive 92/43/EEC (Henry and Aherne, 2014; Whitfield et al., 2011).
In addition to a model evaluation, we include an estimation of the exceedances of CL for the habitats in
the European Natura 2000 network most threatened by N deposition. Moreover, in addressing one of the
objectives of HTAP (Galmarini et al., 2017), we estimated the changes in wet deposition in Europe due to
1) a reduction of global emissions by 20% or to a regional 20% emission reduction solely in 2) North
America or 3) Europe.
The paper is divided into four main sections. Section 2 focuses on the evaluation of model performance
for wet deposition in 2010 (the base case scenario in the context of HTAP and AQMEII3). Section 3
presents the intercomparison of dry deposition. Section 4 provides an overview of the exceedances of the
CL for the most threatened habitats in the Natura 2000 network considering the results of an ensemble,
and finally, Section 5 includes an assessment of the influence of 20% emission reductions alternatively in
Europe, North America and at a global scale on deposition in Europe.





## 2 Model evaluation and intercomparison of wet deposition estimates

### 2.1 Methodology

This Section describes the model simulations (2.1.1), the observations used for model evaluation (2.2.2) and the procedure to evaluate model performance (2.1.3).

Table 1 shows the description and abbreviations of the variables used in the assessment.

### 2.1.1 Model simulations

The simulations for the year 2010 used in this study were carried out using 14 air quality models (Table 2), seven of them as part of AQMEII3, and the other seven models participating in EDT. CHIMERE was involved in both projects, although the model version used in the EDT project is an improved (not yet official) version (Chimere2017b v1.0), and therefore a direct comparison of model results between both simulations (AQMEII3 and EDT) is not possible. More modelling teams than those in Table 2 were involved in the AQMEII3 project, but we kept only those that provided all the variables required for the model performance evaluation in terms of wet deposition, i.e. air concentrations and deposition of related chemical species (except AQ_TR1_MACC, which only provided deposition data). The domain and grid resolution was common for all the models in EDT (except for ED_CMAQ, which used a different domain/projection), with a resolution of 0.25º (lat) × 0.4º (lon). AQMEII3 permitted a more flexible model setup, although outputs had to be produced for a fixed domain with a spatial resolution of 0.25º × 0.25º. Meteorological inputs for the AQMEII3 models were chosen by each participant (Table 2). In EDT, meteorological inputs from the Weather Research and Forecast model (WRF 3.3.1) were provided centrally, although not all models used this common dataset (WRF-Common). In both exercises, boundary conditions were provided to the participants; in AQMEII3 they come from a global model, C-IFS(CB05) (Flemming et al., 2015) running the same scenarios. In EDT boundary conditions come primarily from observations combined with optimal interpolation and long term trends, following the procedure used in the EMEP model (Simpson et al., 2012), with slight adjustments in the context of trend modelling (Colette et al., 2017). Emissions were also fixed in both projects: In AQMEII3 two options were available, Copernicus emissions or HTAP_v2.2 emissions (Janssens-Maenhout, 2015) which for the European region actually contain the Copernicus inventory. In EDT they are ECLIPSE_V5 emissions estimated by the GAINS (Greenhouse gases and Air pollution INteractions and Synergies) model (Amann et al., 2011). More information on the model setups can be found in Galmarini et al. (2017) and Solazzo et al. (2017) for AQMEII3 and Colette et al. (2017) for EDT.

Four simulations were carried out by the AQMEII3 community: a base case (BAS) for 2010; GLO, where emissions were reduced at a global level by 20%; EUR, where emissions were reduced in Europe by 20% and NAM, where emissions were reduced in North America by 20%. Not all the models performed the simulations for all four cases.

### 2.1.2 Observations

Measurements (annual and monthly) made at 88 EMEP monitoring sites for 2010 were provided by the Norwegian Institute for Air Research (NILU), which is the Chemical Coordinating Centre of EMEP,



although not all variables were measured at all sites. A complete description of the monitoring network of
the EMEP program, as well as the sampling methodologies used can be found in Tørseth et al (2012) and
the data are openly accessible from http://ebas.nilu.no/. A summary of sites and variables considered is
included in Table 3 and a map with their location is given in Fig. 1.  Measurements for the gas phase
($HNO_3$, $NH_3$) are quite scarce, which makes it difficult to evaluate models performance for these species.
For example, for annual values, more than two thirds of the sites had measurements for both N and S
deposition and atmospheric $SO_2$ concentrations, while only 10% had data for air concentrations of $HNO_3$
and $NH_3$. More sites than those for $HNO_3$ and $NH_3$ are measuring inorganic aerosols, through these are
analyzed from of PM10 samples in addition to the filterpack which sample both aerosols and gases. One
should be aware that the $NH_4^+$ and $NO_3^-$ concentrations might be underestimated due to the evaporation of
ammonium nitrate. This is the case for both PM10 and filterpack measurements, where the separation of
the nitrogen gases might be biased. The sum of $HNO_3$ and $NO_3^-$, as well as the sum of $NH_3$ and $NH_4^+$ are
however considered unbiased. The filterpack samplers usually have no size cut off, but can be considered
to be around PM10 (EMEP, 2014).
The spatial coverage of the observations used in the evaluation is quite high for most of northern, central
and Western Europe, including Spain, but is quite low in the eastern and southern regions (Fig 1).
**2.1.3 Evaluation**
Model evaluation involved a joint analysis of wet deposition and air concentrations of the corresponding
gas and particle species, as well as precipitation. Accumulated values were considered for precipitation
and wet deposition, whereas mean values were used for air concentrations. Both annual and monthly
values were evaluated. For each model simulation, the following statistics were calculated (Table 4):
normalized mean squared error (NMSE), fractional bias (FB) and the fraction of model estimates within a
factor of two of the observed values (FAC2). The acceptance criteria proposed by Chang and Hanna
(2004; 2005) were used to assess model acceptability:  that is, FAC2 higher or equal to 0.5, values of FB
between -0.3 and 0.3, and NMSE values lower than or equal to 1.5. We define a model as performing
acceptably for a particular variable, when two out of these three criteria are met; in recognition of the
large uncertainties involved in these types of simulations. It should be noted that the acceptability criteria
adopted in this study had their origin in evaluating Gaussian atmospheric dispersion models rather than
photochemical Eulerian grid models. However, due to the absence of established performance criteria for
evaluating modeled atmospheric deposition, these criteria were nevertheless adopted in this study while
future work may be directed at developing performance goals more specifically tailored towards
atmospheric deposition. To illustrate model performance for each variable, the three assessment statistics
are shown on the same graph by plotting NMSE against FB and using a different symbol to indicate
whether a model meets the acceptance criterion of Chang and Hanna (2004) for FAC2 (FAC2 $\geq$ 0.5).
These plots include shaded areas that correspond to areas meeting the acceptance criteria of Chang and
Hanna (2004) (blue for NMSE, red for FB). In addition, the theoretical minimum NMSE for a given value
of FB is also plotted (parabolic dashed lines) (Chang and Hanna, 2004). These "smile plots", as they are
called hereafter, were produced considering annual and monthly data,  and also by month, in order to
illustrate the seasonal behavior. All statistics were calculated in two ways: 1) independently for each



variable, so as to have the largest number of available sites for each variable, and 2) considering a
common set of sites for wet deposition and air concentrations of the respective gas and particle species for
each deposition type: oxidized nitrogen (ON), reduced nitrogen (RN) and sulfur (S).
Additional statistics, (mean gross error, MGE, normalized mean bias, NMB, normalized mean gross error,
NMGE, root mean squared error, RMSE, correlation coefficient, r, coefficient of efficiency, COE and
index of agreement, IOA), were also calculated, as defined in the Auxiliary material (AM 3.9).
In order to provide robust estimates of N and S deposition and their uncertainties for further applications,
such as the one in Section 4, a multi-model ensemble was constructed using the mean and standard
deviation of the total deposition for each grid cell calculated from the estimates of the best performing
models. A given model was included if it met at least two of the three acceptability criteria for wet
deposition, gas and particle concentration, considering results for all the available sites and common sites.
The main problem with this approach was that gas concentrations of NH3_N and HNO3_N were only
measured at a few measurements sites. When the criteria for these gas pollutants were the only ones
failing, we retained the model (ED_EMEP, AQ_FI_MACC&HTAP) if the criteria for total concentrations
was met (note that TNO3 and TNH4 were measured at some sites where no separate measurements of gas
and particle air concentrations were made and thus model performance for these variables as well as
TSO4 was only evaluated for all available sites).

## 2.2 Results and discussion

The evaluation statistics for the selected models are provided in the tables in AM 3.6. These results are
represented visually in the *smile plots* of Fig. 2 (based on annual values, considering all the available sites
for each variable) and AM 3.1 (based on monthly values), which also show the degree to which the
acceptability criteria were met for all models. Fig. 3 shows the *smile plots* considering only the common
set of sites (sites with measurements of all the variables), to facilitate the analysis with regards to the
interdependencies of model performance for different variables. Results for the ensemble, calculated as
exposed in Section 2.1.3 are also included in smile plots and tables, in order to have a view of the quality
of its performance. Considering the criteria in Section 2.1.3 and tables AM 3.7 (calculated for all the
available sites) and 3.8 (for common sites) jointly (that is, the criteria had to be met in both tables, on an
annual basis), the ensemble was composed of AQ_DK1_HTAP, ED_CHIM, ED_EMEP, ED_LOTO,
AQ_FI1_MACC, AQ_FI1_HTAP and ED_MATCH for N deposition (considering both ON and RN at
the same time; gridded information for AQ_UK1_MACC and AQ_UK2_HTAP, passing the acceptance
criteria, was not available). For S deposition the models meeting the criteria for SO2_S, PM_SO4_S and
WSO4_S were ED_EMEP, ED_LOTO, ED_MATCH, AQ_FI1_HTAP, AQ_FI1_MACC and
AQ_UK1_MACC (AQ_UK1_MACC gridded information was not available for all the variables, so it
was not included in the ensemble). Figs. 4 and 6 show the deposition of N and S for the selected models
and the ensemble. The ensemble was calculated to facilitate the analysis in Section 4. Maps of annual wet
deposition for all the models are shown in AM 1. Other criteria to select the models in the ensemble or the
way to calculate it would lead to a different ensemble.



Accumulated precipitation was also evaluated. In general, monthly and annual precipitation rates
estimated by the models agree reasonably well with the observations. The *smile plots* for precipitation in
Fig. 2 and AM 3.1 (and the tables in the AM 3.6) show that all the models meet all acceptability criteria,
with the exception of AQ_DE1_HTAP, which narrowly misses the FB criterion for this variable.
AQ_FRES1_HTAP had the lowest errors (NMSE) and the highest correlation with the observed
precipitation values (r).
In the case of WNO3_N (abbreviations in Table 1) a large variability was found (AM 1.2), with
ED_MINNI and AQ_DE1_HTAP giving the lowest values and AQ_TR1_MACC giving the highest. The
*smile plot* in Fig. 2 (also included in AM 1.2 to facilitate interpretation) and tables in AM 3.6 show that
the models tended to underestimate the observed WNO3_N, (ED_EMEP and AQ_DK1_MACC very
slightly underestimating), on average, with the exception of AQ_TR1_MACC and ED_MATCH, that
overestimated slightly. The results for ED_MINNI are consistent with the study by Vivanco et al. (2016),
who evaluated several models (EMEP, CHIMERE, LOTOS-EUROS, MINNI, CMAQ and CAMX) for
four one-month campaigns during 2006, 2007, 2008 and 2009. Most of the models meet at least two of
the three acceptability criteria for both monthly and annual wet deposition values, with the exceptions
being ED_MINNI and AQ_DE1_HTAP, which substantially underestimated deposition.  As shown in
AM 3.6 all the models performed acceptably for TNO3_N, except AQ_DE1_HTAP for the monthly data
and ED_CMAQ for the annual data. Interestingly, all the models performed worse for atmospheric
concentration of the gaseous form (HNO3_N) than for the particulate form (PM_NO3_N) (also visible in
Fig. 3), with no model performing acceptably for the monthly data. Boxplots in AM 4 indicate an
underestimation of the HNO3:TNO3 ratio in winter for most of the models. The *smile plots* in the AM 3.2
also show the highest errors and underestimation of HNO3_N during these months. In fact, no model
meets two criteria in Jan, Feb, Mar, Nov and Dec for this pollutant. Most models overestimated HNO3_N
in the period May-Sep, with the exception of July for which the models tended to underestimate
concentrations. This summer period was also when the models estimated the highest HNO3:TNO3 ratios,
many of which were higher than observed (especially for AQ_FRES1_HTAP, ED_MINNI). The models
performed best for the gaseous component during Jun-Aug. Most models underestimate both WNO3_N
and HNO3_N and overestimate PM_NO3_N for the winter period (Oct-Mar), which could suggest a too
efficient gas-to-particle conversion during these months in some cases, with maybe low deposition
efficiency for the particle phase. In the case of AQ_DE1_HTAP the underestimation of deposition, as
well as gas and particle air concentration could be related to an underestimation of $NO_2$ or HNO3 (via a
low $NO_2$ to $HNO_3$ conversion rate). ED_EMEP overestimates WNO3_N and PM_NO3_N, but
underestimates HNO3_N (according to annual values for common sites in AM 3.8), which could be
related to a too high gas deposition.
For WNH4_N there were also large differences between the models giving the lowest values
(AQ_DE1_HTAP, AQ_FRES1_HTAP and ED_MINNI), and the models giving the highest
AQ_TR1_MACC). Most of the models meet at least two of the three acceptability criteria for this
pollutant, with the exceptions being AQ_DE1_HTAP, AQ_FRES1_HTAP and ED_MINNI. Similar to
WNO3_N, Fig. 2 (also included in AM 1.1) and tables in AM 3.6 show that the models tended to
underestimate WNH4_N, with the exception of AQ_TR1_MACC and ED_MATCH. However, unlike



WNO3_N, this underestimation seems to correlate with an overestimation of the gaseous form (NH3_N)
on an annual basis (except for ED_EMEP, which has a very low bias for both pollutants and
ED_MATCH, which overestimates WNH4_N slightly). This is likely due to an underestimation of wet
removal processes for the gas phase, but it can also be related to other issues, such as a general
underestimation of NH3 dry deposition or an overestimation of emissions or even to measurement
locations far from agricultural sources of ammonia and therefore not representative of the grid square.
The overestimation of NH3_N mainly occurs in autumn and winter (Jan, Feb, Nov, Dec), as can be
inferred from the monthly smile plots of NH3_N in the AM 3.3, which shows a poorer model
performance for this period (no model meets all three criteria). It is interesting to see that this
overestimations of NH3_N during Nov-Jan takes place when HNO3_N is underestimated, which could
indicate an excessive conversion of HNO3 to particle due to an excess of $NH_3$ (aerosol nitrate may be
formed if enough ammonia is available) and favored with low temperatures. Ammonium is quite well
reproduced, with all the models meeting the acceptance criteria both on an annual basis and a monthly
basis. All in all, tables in AM 3.6 indicate a general underestimation of wet deposition for reduced
nitrogen, with a tendency to overestimate TNH4. There is more variability between the model estimates
of the NH3:TNH4 ratios for the winter months (AM 4) with the EDT models estimating lower ratios. It
should be noted that some models do not distinguish between precipitation types and use the same
scavenging rates for snow and rain, which could lead to substantial differences between model results.
Substantial differences were also found for WSO4, from the lowest values for ED_CHIM up to the
highest for AQ_TR1_MACC and ED_MATCH. Most of the models meet at least two of the three
acceptability criteria for WSO4, apart from AQ_DK1_HTAP, AQ_FRES1_HTAP, ED_CHIM and
ED_MINNI. Similar to the N deposition, the models tended to underestimate the observed values (Fig. 2),
with the exception of AQ_TR1_MACC, AQ_UK2_HTAP, ED_EMEP and ED_MATCH. The tendency
to underestimate WSO4_S by most models, and similarly to the reduced nitrogen, is overall occurring
simultaneously with an overestimation of the gaseous pollutant (SO2_S) on an annual and monthly basis.
As shown in the monthly *smile plots* in the AM 3.4, the models generally underestimate WSO4_S for all
months although the bias tends to be smaller (and even positive for some models) during the winter
period (Nov-Feb). The bias for SO2_S does not have a seasonal cycle and the largest errors occur in Mar,
Jun and Nov. Model performance is generally better for the particulate concentrations (PM_SO4_S)
although some large errors occur in the winter (Nov-Jan). All models tended to overestimate TSO4, with
the exception of ED_CHIM, ED_EMEP and ED_LOTO, and most models also tended to overestimate the
SO2:TSO4 ratios.
In summary, and considering the whole picture, wet deposition fluxes are generally underestimated for
WSO4_S and WNH4_N, and in winter in the case of WNO3_N. There are indications that the aqueous
and heterogeneous chemistry (e.g. those involving conversion of NOx to HNO3) could be too slow or
under-represented in the models, especially in winter, evidenced by an overestimation of primary gaseous
pollutants, especially NH3 and SO2 for this period and an underestimation of the secondary pollutant
HNO3 (also formed via heterogeneous chemistry). However, this behavior (simultaneous overestimation
of NH3_N and underestimation of HNO3_N in winter) could also be due to an excessive formation of
nitrates (favored by low temperatures) due to a potential excess of NH3 (aerosol nitrate may be formed





only if enough ammonia is available). This excess NH3 could be due to an overestimate of NH3
emissions during these months. The fact that sulphate concentration is also low for several models in Jan
and Feb and SO2 somewhat high could be due to an underestimate of the conversion to aerosol (sulphate)
via aqueous chemistry, which could be another cause of the excess NH3.
**3 Model intercomparison of dry deposition**
Figures in AM 2 show maps of dry deposition for oxidized nitrogen (OND) (AM 2.2), reduced nitrogen
(RND) (AM 2.1), total N (ND) (AM 2.4) and S (AM 2.5). Unfortunately, not all the models participating
in AQMEII3 provided the complete set of outputs, and therefore it was not possible to study the estimated
dry deposition for all of them. Maps of dry deposition of total N (ND) for all the models show the highest
values over France, Germany and other areas in the center of the domain. Differences between models
can be seen in both high and low emission areas. Models have different deposition algorithms and, even
when similar, they can have different input, such as land use or the leaf index area. It would be interesting
in future studies to analyse how much different these parameters in the models are, due to their relevant
importance in dry deposition estimates. The highest values of dry deposition for total N (AM 2.4) are
found for ED_CMAQ, with values higher than 1900 mg N m$^{-2}$ (annual accumulated value) over large
areas in the central and western parts of the domain and mainly due to the contribution of the oxidized
species. AQ_FRES1_HTAP estimated the lowest values whereas the rest of model estimates have more
similar spatial patterns. Significant differences can be found when looking at the gas and particle
deposition for the AQMEII3 participants. Two gases, NO2 and HNO3 can contribute to OND. As can be
inferred from AM 2.3, AQ_DK1_HTAP estimate the main contribution from the gas phase, whereas in
the case of AQ_TR1_MACC, highest contributions to OND come from the particle phase. This highlights
the importance of making measurements that can shed more light on these processes, providing modelers
with data that can be used to parameterize and evaluate the different processes. For RN only
AQ_FRES1_HTAP, AQ_UK2_HTAP and AQ_FI1* in AQMEII3 provided the information required to
calculate RND. The models estimate similar spatial distributions of RND, with the highest values in the
Netherlands, the western part of France, Denmark and Belgium, as well as some high values in the area of
the Alps. Spatial distributions are also similar for dry deposition of S (AM 2.5; higher values mainly over
Poland, The Netherlands, United Kingdom, Germany and Southeastern Europe), although in this case
with higher differences in values, as it can be inferred from maps in AM 2.5. ED_CMAQ presents a
different spatial pattern, with high values also over sea, due to the consideration of sulfates coming from
sea salt in this model application.

**4 Deposition of N over areas in Nature 2000 network**
In this section, we first analyze the representativeness of the monitoring sites used in the evaluation of
model deposition with a focus on habitat conservation. Secondly, the estimated deposition by the multi-
model ensemble is used to evaluate the total N deposition (dry + wet) to the protected habitats. Finally, a
simple evaluation (where possible) of the CL exceedances is presented. Together with S deposition, N
deposition also contributes to acid deposition. However, as mentioned in the introduction, only 5% of the



Natura 2000 area was at risk of acidification in 2010 and so the focus of this part of the study is on the
exceedances of CLs for the nutrient N.

**4.1. Representativeness of monitoring sites for conservation purposes**
The EMEP measurements are regional representative (Tørseth et al 2012 , EMEP, 2014) and have
historically been considered to represent an area larger than the size resolution of the EMEP atmospheric
dispersion model (for the grid with 50x50km2 of horizontal resolution). This resolution was taken as a
reference for establishing a buffer zone of 2500 km2 around the receptors. The protected habitats inside
the buffer zone were determined by intersecting the surface area of the Natura 2000 network (EEA,
2017), with the cover of the most-likely habitats in Europe using EUNIS level-1 classification (EEA,
2015). Previously to this, aquatic, aquatic-related and anthropic habitats (such as gardens or arable lands)
were excluded, in order to study only natural and semi-natural terrestrial ecosystems. The surface area
covered by each habitat class included in the Natura 2000 network was plotted against the surface area of
the same protected habitat classes within the above-mentioned buffer zones, in relative values with
respect to their respective totals (Table 5, Fig. 8). The most represented terrestrial habitats in the entire
network are broadleaved deciduous woodland, coniferous woodland, mesic grasslands and mixed
deciduous and coniferous woodland (EUNIS classifications G1, G3, E2 and G4, respectively). The results
indicate that the selected monitoring sites represent the main classes of terrestrial habitats fairly well, with
G4 deviating most, with an overrepresentation of 51% within the protected buffered area with respect to
the entire Natura 2000 network.
The same exercise was performed using only monitoring sites measuring all N species (including in
precipitation, gaseous and particulate N). Only 8 monitoring sites, distributed between the United
Kingdom, Switzerland and Eastern Europe, have the complete set of N pollutant measurements. Since the
Natura 2000 network has no presence in Switzerland, only 6 sites could be evaluated for
representativeness. Among the most represented habitats, G1 and G3 deviated the most in their
representation. In any case, this subset can be considered small and poorly distributed across Europe.
Therefore, the evaluation of model results for total concentration and deposition of N pollutants in Europe
is still far from being representative in terms of conservational purposes.
**4.2. Risk assessment of atmospheric N deposition in the Natura 2000 network**
The mean and standard deviation (SD) for total deposition of N obtained from the ensemble model were
combined with revised empirical CL (Bobbink and Hetteling, 2011) to provide a risk assessment of N
deposition effects on vegetation in the Natura 2000 network. This evaluation constitutes a first approach,
which helps to locate the most-likely areas and major terrestrial habitat classes at risk of eutrophication as
a result of atmospheric N deposition. Further research (particularly on habitat specific CL) and a wider
monitoring network (particularly to evaluate models' performance for dry deposition) are needed to carry
out a more accurate risk assessment. It is also interesting to bear in mind that even though recent studies
(e.g. Cape et al., 2012; Izquieta-Rojano, 2016; Matsumoto et al., 2014) have highlighted the important
contribution of the organic form to total N deposition (from 10 to more than 50%), there are still
important gaps in our knowledge of the role of organic fraction in the N cycle and scarce attempts to



include it in the measurement networks (e.g. Walker et al., 2012). Deposition of dissolved organic N
constitutes another variable involving uncertainty in the actual understanding of the N cycle (Izquieta-
Rojano et al., 2016) and, consequently, in the risk assessment of N deposition. Further research is
therefore needed to understand the role that organic N plays in ecosystem functioning, biogeochemical
cycles and even human health.

Ensemble deposition maps were projected and resampled to coincide with the EUNIS habitat grid (level 1
classification; ETRS89 LAEA projection; 100 m ×100 m cell size). The mean±SD values were used as
estimates of lower and upper uncertainty limits for the deposition, which were then compared to the mean
CL attributed to each habitat class (Table 5; based on those from Bobbink and Hetteling, 2011). Those
areas in which the class-attributed CL was exceeded by any of the values (mean-SD; mean; mean+SD)
were identified. The area presenting exceedances of empirical CL ($CL_{exc}$) was summed for each EUNIS
level-1 habitat class (Table 5). The areas showing $CL_{exc}$ were mapped for the most threatened habitat
classes (Fig. 9). In the case of similar habitats with similar distributions, a joint map is shown (D1 and
D2; G3 and G4). Values of $CL_{ex}$ in Fig. 10 indicate the area exposed to an exceedance of the CL
expressed as percentage of the total area evaluated for each particular habitat class. These values were
also calculated considering the total deposition of N from AQ_FI_MACC, as this model was used to
estimate the variation in deposition due to changes in emissions, as it will be later explained. All these
operations were performed using ArcGIS 10.2 (ESRI, Redlands CA, USA).
The six habitats with the largest surface area with a mean ensemble deposition above their respective CL
were "alpine and subalpine grasslands" (E4), "coniferous woodlands" (G3), "mixed deciduous and
coniferous woodlands" (G4), "raised and blanket bogs" (D1), "artic, alpine and subalpine scrub" (F2) and
"valley mires, poor fens and transition mires" (D2), with critical load exceedances covering 65%, 34%,
32%, 24%, 16% and 11% of their respective areas (Table 5). Alpine and subalpine grasslands were also
detected as the types most jeopardized by N deposition, in a similar study for Spanish protected areas
using 2008 simulations from EMEP and CHIMERE models (García-Gómez et al., 2014). These habitats
are usually located in areas with complex topography, where model estimates of atmospheric deposition
can be more spatially inaccurate, as suggested in previous studies (e.g. García-Gómez et al., 2014;
Simpson et al., 2006). The scarcity of monitoring sites at high altitude to evaluate model simulations can
be considered as a major uncertainty in the risk assessment for N deposition.
The variation among the models included in the ensemble, represented here by the standard deviation
(SD) of the ensemble, mostly affected E4 (Table 5). The reduction of the area at risk of this habitat class
is remarkable high (-50%), when the lower limit of the deposition is used (mean-SD; Table 5). This might
indicate that the CL is exceeded in most areas by a narrow margin. Within the other five habitat classes
with the highest $CL_{exc}$ area, the area at risk decreased by 13% and increased by 16% on average, when the
lower and upper limits of deposition are used. These same six habitats were again found to present the
largest areas showing $CL_{exc,}$ when using AQ_FI1_MACC estimates, although some differences were
found (seen Figure 10).
Apart from the uncertainty in modelled deposition, the uncertainty in the CL attributed to the habitat
classes should also be considered. On the one hand, some CL proposed in the CLRTAP revision are based



on expert judgment (e.g. those for E2, F5 or G4) and some were averaged from those proposed for several
subclasses (e.g. for E1 and F4). On the other hand, even when the proposed CL are reliable and match
perfectly with the habitat classes evaluated in this study, an adjustment linked to more local conditions is
recommended (e.g. for D1 it is recommended to vary the applied CL as a function of the precipitation
range or the water table level). However, since a CL averaged from the proposed range was used for each
habitat class and the evaluation was performed on a broad scale, we consider that the results are suitable
for the purpose of this work, which is highlighting the protected areas and terrestrial habitats with the
highest probability of suffering eutrophication. Finally, the use in this approach of a modelled dry
deposition that is in fact weighted for the different land use inside each grid cell might lead to an
underestimation of, for instance, forests risks, as the dry deposition for plant surfaces is higher than for
other land uses, and it is currently smoothed during the weighting process. To perform a more accurate
assessment, habitat-type-specific values for dry deposition of N are necessary. It is, therefore,
recommended that chemical transport models provide dry deposition data as a function of leaf area index
(LAI) or habitat type in order to be more suitable for risk assessment studies.

## 409    5 Contribution to N and S deposition in Europe of different regions (NA, EU, GLO)

### 410    5.1 Methodology

As we have previously described in the framework of AQMEII3 activities, and to give scientific support
to the HTAP task force, research activities have included an evaluation of the influence of a reduction of
emissions in some parts of the Northern Hemisphere on the air quality other regions. Along these lines,
some models ran simulations with 1) a 20% reduction of global emissions (GLO), 2) a 20% reduction of
emissions in Europe (EUR) and 3) a 20% reduction of emissions in North America (NAM). According to
the acceptance criteria described in Section 2, and the availability of models running the different
emission scenarios, we chose AQ_FI1_MACC as a representative model to demonstrate the effects of the
different emission reduction scenarios. For WNO3 the results from the AQ_FRES1_HTAP model were
included as well, as this model performed acceptably for this pollutant and simulated the three
perturbation scenarios.
The effect of each scenario was calculated in terms of deposition (mgN/m2) and percentage changes with
respect to the base case (%). Differences between the base case simulation (no emission reduction) and
the different scenarios were calculated for wet and dry deposition of ON, RN and S, as well as for total
deposition of N and S.

### 426    5.2 Results

Maps reflecting the effect of the reduction of 20% of emissions in the different scenarios are included in
figures 11 and 12, for total N and S (including both oxidized and reduced N, as well as wet and dry
deposition), in absolute and relative terms. In general, a 20% reduction of total N and S deposition is
found when global emissions are reduced by 20% (although somewhat lower for N in the United
Kingdom, the Netherlands and in Belgium). When a 20% emission reduction is only applied in Europe,





the deposition of N and S is decreased by 10-20%. When emissions are reduced in North America only,
deposition at the eastern areas of the domain is reduced by about 2%, (Fig. 9). Im et al. (2017) found also
an almost linear response to the change in emissions for $NO_2$ and $SO_2$ air concentration, for the global
perturbation scenario, with slighter smaller responses for the European perturbation scenario and very
small influence of the long-range transport, noticeable close to the boundaries.
Similar maps for wet and dry deposition are presented in AM 5 and AM 6, for wet and dry deposition.
For WNO3_N the global emission reductions have the largest effect on European deposition, with the
largest changes in wet deposition in the Alpine area (North Italy, Southern Germany). These areas are
also affected in terms of WNH4_N, although in this case the emission reduction affects larger areas in
Germany and The Netherlands. For WSO4_S (AM) the highest impacts are found on the Balkan
Peninsula, especially the south of Bulgaria, Rumania and Serbia. These quantities represent a reduction of
about 20% of the base case deposition in most parts of Europe, even a bit higher for WNO3_N in the
Alpine area according to AQ_FI1_MACC. For AQ_FRES1_HTAP the reduction for WNO3_N is lower,
in the range 14-20% for the whole domain.
When emission reductions only occur in Europe, the changes in wet deposition are somewhat lower than
for a global reduction according to AQ_FI1_MACC, (AM 5.1, AM 5.2). Reductions in WNH4_N are
similar to those of the global emission reduction scenario in western and central Europe, but substantially
smaller in the eastern and northern parts of the domain, which are influenced more strongly by non-
European emissions to the east.  Larger differences are found between the global and European emission
reduction scenarios for WNO3_N, with an influence of non-European emissions that extends throughout
the domain. In many countries wet deposition decreases by about 10% for the European emission
reduction scenario, and a 20% reduction is only found over some central areas.  The situation is similar
for  WSO4_S,  albeit  with  even  larger  contributions  from  non-European  emissions.  For
AQ_FRES1_HTAP, the reduction of WNO3_N is similar to that estimated by AQ_FI1_MACC, although
the range of reduction is smaller. Emission reductions in NA have a very small effect on European wet
deposition (around a 1-2%), with reductions mostly concentrated in the western part of the domain
(Iceland, Ireland, United Kingdom, Portugal, France, Spain, Norway. This pattern is also reproduced by
AQ_FRES1_HTAP, although the absolute changes for AQ_FI1_MACC are larger in the central area and
smaller on the Iberian Peninsula. The effect of global emission reductions on dry deposition is similar to
that for wet deposition, although the relative reductions are slightly smaller for DNO3_N (except in the
east and south of the domain) and slightly larger for DNH4_N and DSO4_S than for WNO3_N,
WNH4_N and WSO4_N, respectively (AM 5, AM 6).  The differences between the relative changes in
wet and dry deposition are similar for the European emission reduction scenario, although the relative
change is larger for the dry deposition in the east of the domain. The influence of emission reductions in
NA on the wet deposition is generally larger than that on the dry deposition.
Differences between the global emissions reduction scenario and the European emission reduction
scenario, discounting the effect of NAM, indicate that there is an influence of emissions from other
regions, especially to the east of the domain that could produce a 10% reduction in deposition over certain
areas. This is in agreement with results from studies carried out within the framework of the HTAP task



force using global models, which estimate that 5-10% of European N deposition is the result of non-
European emissions (Dentener et al., 2011; Sanderson, 2008).
We also estimated how much these reductions in emissions affected the risks of N impacts in the Natura
2000 areas. As can be inferred from Figure 10, there is a significant reduction in the habitat area
withstanding CLexc for the scenarios GLO and EUR, compared with the base case (AQ_FI1_MACC).
Particularly, the most jeopardized habitat types showed a reduction of more than a third in their overall
threatened area. Both reduction scenarios showed almost similar values of CLexc, with only slight
differences in E4 (where GLO reduction produces a slightly larger decrease in CLexc). G3 and G4
habitats are the most affected, for which the exceeded area was approximately halved as a result of the
emission reduction. In the case of NAM, no decrease is observed, indicating the low impact of
hemispheric transport from North America to Europe, at least in terms of N deposition in 2010.6

## 6 Conclusions

A comparison of the wet and dry deposition of N and S estimated by 14 air quality models participating in
the projects AQMEII3 and EURODELTAIII revealed considerable differences between the models. An
evaluation of model performance was carried out, jointly considering air concentrations and wet
deposition of the relevant compounds. Very few measurements of gaseous species (HNO3 or NH3) were
available, making it difficult to do a fair and complete evaluation. In general, most of the models meet at
least two of the three acceptability criteria (NMSE < 1.5, |FB| < 0.3, FAC2 > 0.5) for both monthly and
annual wet deposition values, with the exceptions of ED_MINNI and AQ_DE1_HTAP, which
substantially underestimated deposition. All the models performed acceptably for TNO3_N, except for
AQ_DE1_HTAP for the monthly data and ED_CMAQ for the annual data. All the models performed
worse for atmospheric concentrations of the gaseous form (HNO3_N) than for the particulate form
(PM_NO3_N), with no model performing acceptably for the monthly data, and most models
underestimating the HNO3:TNO3 ratio during the winter months. It is however important to note that the
observations of independent NO3- and HNO3 are not measured with an unbiased method (same as NH3
and NH4+), so it is difficult to draw strong conclusions of the model performance for these compounds.
For WNH4_N, there was a general underestimation, that seems to correlate with an overestimation of the
gaseous form (NH3_N) on an annual basis (except for ED_EMEP, which has a very low bias for both
pollutants, and ED_MATCH, which overestimates WNH4_N slightly) mainly as a result of model
estimates for autumn and winter (Jan, Feb, Nov, Dec). Similarly, to the reduced nitrogen, most models
tend to underestimate wet deposition of sulfur (WSO4_S) and overestimate the gaseous pollutant
(SO2_S) on an annual and monthly basis.
Large differences were found between the dry deposition estimates of the models, highlighting the
importance of obtaining measurement data to evaluate model performance. This point is important,
considering the significant contribution of dry deposition to total deposition.
A multi-model ensemble was constructed using the better-performing models for wet deposition (N and
S) and having also estimated dry deposition. For N, the ensemble was produced as the mean of
AQ_FI1_MACC, AQ_FI1_HTAP, AQ_DK1_MACC, ED_EMEP and ED_MATCH models, and was
used to calculate exceedances of empirical critical loads for nitrogen for habitats in the European Natura



2000 network. Six habitats were identified as having critical load exceedances covering more than 10% of their total area: "alpine and subalpine grasslands" (E4), "coniferous woodlands" (G3), "mixed deciduous and coniferous woodlands" (G4), "raised and blanket bogs" (D1), "artic, alpine and subalpine scrub" (F2) and "valley mires, poor fens and transition mires" (D2), with critical load exceedances covering 60%, 30%, 29%, 22%, 13% and 10% of their respective areas. The variation among the ensemble models, in terms of the standard deviation of the ensemble, mostly affected E4, with 85% of the habitat area exceeded for the upper deposition estimate. It's important to point out that in addition to the uncertainty in modelled deposition, the CL attributed to a given habitat is also uncertain. Extending the deposition monitoring networks in European mountains would be not only beneficial for the study of atmospheric deposition, but also for model evaluation and risk assessment for these particularly threatened areas.

The reduction of 20% of emissions at global scale produces a 20% of reduction in total deposition of N and S, with the main contributor being Europe, according to the estimates of A_FI1_MACC model. This reduction of total deposition is directly related to a decrease of the CLexc found for the different habitats in Natura 2000 network, especially for G3 and G4, for which the exceeded area was approximately halved as a result of the emission reduction. Hemispheric transport of air pollutants from NAM has a low impact on wet deposition, mostly concentrated over the Atlantic area.

**4 Acknowledgements**

CIEMAT work has been financed by the Spanish Ministry of Agriculture and Fishing, Food and Environment. The MATCH participation was partly funded by the Swedish Environmental Protection Agency through the research program Swedish Clean Air and Climate (SCAC) and NordForsk through the research programme Nordic WelfAir (grant no. 75007). The views expressed in this article are those of the authors and do not necessarily represent the views or policies of the U.S. Environmental Protection Agency

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





**Table 1**: Abbreviation used in this publication

| | |
|---|---|
| Wet deposition of oxidized N | WNO3_N |
| Wet deposition of reduced N | WNH4_N |
| Wet deposition of S | WSO4_S |
| Dry deposition of oxidized N | DNO3_N |
| Dry deposition of reduced N | DNH4_N |
| Dry deposition of S | DSO4_S |
| Atmospheric concentration of N from nitric acid | HNO3_N |
| Atmospheric concentration of N from nitrate in $PM_{10}$ | PM_NO3_N |
| Total oxidized N concentration = $HNO_3$_N + PM_NO3_N | TNO3_N |
| Atmospheric concentration of N from ammonia | NH3_N |
| Atmospheric concentration of N from ammonium in $PM_{10}$ | PM_NH4_N |
| Total reduced N concentration = $NH_3$_N + $PM\_NH_4$_N | TNH4_N |
| Atmospheric concentration of S | SO2_S |
| Atmospheric concentration of S from sulphate in $PM_{10}$ | PM_SO4_S |
| Total S concentration = SO2_S + PM_SO4_S | TSO4_S |
| Precipitation | PRECIP |






Table 2 Meteorological and CTM model used by each participant.

|  | AQMEII3 | | | EDT | |
|  | METEO | CTM |  | METEO | CTM |
| --- | --- | --- | --- | --- | --- |
| AQ_DE1_HTAP | COSMO-CLMy | CMAQ (v4.7.1) | ED_CHIM | WRF-Common* | CHIMERE (Chimere2017b v1.0) |
| AQ_DK1_HTAP | WRF | DEHM | ED_CMAQ | WRF-Common (adapted to different projection ) | CMAQ (v5.0.2) |
| AQ_FI1_HTAP/_MACC | ECMWF | SILAM | ED_EMEP | WRF-Common | EMEP (rv4.7) |
| AQ_FRES1_HTAP | ECMWF | CHIMERE (vchim2013) | ED_LOTO | RACMO2 (nudged) | LOTOS (v1.10.005) |
| AQ_UK1_MACC | WRF | CMAQ (v5.0.2) | ED_MATCH | HIRLAM | MATCH (VSOA April 2016) |
| AQ_UK2_HTAP | WRF | CMAQ (v5.0.2) | ED_MINNI | WRF-Common | MINNI (V4.7) |
| AQ_TR1_MACC | WRF | CMAQ (v4.7.1) |  |  |  |

- Colette et al. 2017





**Table 3:** Number of sites for each pollutant

| | | | |
|---|---|---|---|
| WNO3: 59 | TNO3: 45 | HNO3: 12 | PM_NO3: 32 |
| WNH4: 61 | TNH4: 39 | NH3: 12 | PM_NH4: 27 |
| WSO4: 61 | SO2: 57 | TSO4: 18 | PM_SO4: 21 |


**Table 4:** The three metrics relating modelled concentrations (M) with the observed values (O), used for evaluating
**model performance.**

| NMSE | $NMSE = \dfrac{\overline{(O-M)^2}}{\overline{O}\ \overline{M}}$ | <= 1.5 |
|---|---|---|
| FB | $FB = \dfrac{2(\overline{M}-\overline{O})}{(\overline{O}+\overline{M})}$ | $|FB| <= 0.3$ |
| FAC2 | Fraction of model estimates within a factor of two of the observed values $$0.5 \le \frac{M}{O} \le 2.0$$ | FAC2 >= 0.5 |





**Table 5.** Coverage representation, mean ensemble deposition a critical load exceedance for major terrestrial habitat classes within the Natura 2000 network.

| Habitat group | EUNIS code | Habitat class | Natura 2000 [a] | Receptors [b] | Avg. Dep (kgN/ha) [c] | CL (kgN/ha) [d] | CL_exc [e] | CL_exc (Dep.-SD) [f] | CL_exc (Dep.+SD) [f] |
|---|---|---|---|---|---|---|---|---|---|
| Peatlands | D1 | Raised and blanket bogs | 1.9% | 2.9% | 5.98 | 7.50 | 24% | 13% | 37% |
| | D2 | Valley mires, poor fens and transition mires | 0.2% | 0.1% | 6.94 | 12.50 | 11% | 7% | 16% |
| | D3 | Aapa, palsa and polygon mires | 2.1% | 1.1% | 1.49 | | | | |
| | D4 | Base-rich fens and calcareous spring mires | 0.1% | 0.1% | 9.02 | 21.25 | 1% | 0% | 2% |
| | D5 | Sedge and reedbeds | 0.5% | 0.3% | 8.05 | | | | |
| | D6 | Inland saline and brackish marshes and reedbeds | < 0.1% | < 0.1% | 11.34 | | | | |
| Grasslands | E1 | Dry grasslands | 0.5% | 0.1% | 5.41 | 15.75 | 0% | 0% | 0% |
| | E2 | Mesic grasslands | 14.1% | 9.8% | 9.02 | 20.00 | 2% | 1% | 3% |
| | E3 | Seasonally wet and wet grasslands | 1.8% | 0.8% | 8.83 | 16.25 | 5% | 2% | 10% |
| | E4 | Alpine and subalpine grasslands | 1.3% | 1.3% | 8.40 | 7.50 | 65% | 15% | 85% |
| | E6 | Inland salt steppes | 0.5% | 0.1% | 7.60 | | | | |
| | E7 | Sparsely wooded grasslands | 1.3% | 0.4% | 5.24 | | | | |
| Shrublands | F2 | Arctic, alpine and subalpine scrub | 2.7% | 3.9% | 5.07 | 10.00 | 16% | 5% | 32% |
| | F3 | Temperate and Mediterranean-montane scrub | 3.6% | 3.1% | 4.25 | | | | |
| | F4 | Temperate shrub heathland | < 0.1% | < 0.1% | 4.67 | 15.00 | 0% | 0% | 1% |
| | F5 | Arborescent and thermo-Mediterranean brushes | 2.7% | 2.4% | 6.11 | 25.00 | 0% | 0% | 0% |
| | F6 | Garrigue | 0.6% | 1.1% | 6.39 | | | | |
| | F7 | Spiny Mediterranean heaths | 1.1% | 1.1% | 5.72 | | | | |
| | F8 | Thermo-Atlantic xerophytic scrub | 0.3% | 0.0% | nd | | | | |
| | F9 | Riverine and fen scrubs | < 0.1% | < 0.1% | 4.15 | | | | |
| | FB | Shrub plantations | 0.8% | 0.3% | 7.63 | | | | |
| Woodlands | G1 | Broadleaved deciduous woodland | 25.1% | 23.4% | 8.50 | 15.00 | 4% | 1% | 14% |
| | G2 | Broadleaved evergreen woodland | 1.2% | 0.4% | 6.88 | 15.00 | 0% | 0% | 5% |
| | G3 | Coniferous woodland | 20.7% | 25.6% | 7.83 | 10.00 | 34% | 14% | 53% |
| | G4 | Mixed deciduous and coniferous woodland | 9.4% | 14.2% | 8.61 | 10.75 | 32% | 13% | 58% |
| | G5 | Early-stage woodland and semi-natural stands | 7.6% | 7.5% | 6.16 | 7.50 | | | |

a) representation within the Natura 2000 network; b) representation within the Natura 2000 network in the joint of the buffered areas; c) weighted mean of N deposition for each habitat class according to ensemble results; d) attributed critical load in this work (based on empirical critical loads from Bobbink and Hetteling, 2011); e) area withstanding an exceedance of the CL, expressed as percentage of the total area evaluated for each particular habitat class; f) area withstanding an exceedance of the CL, when using an ensemble deposition value of mean minus/plus the standard deviation of the ensemble mean





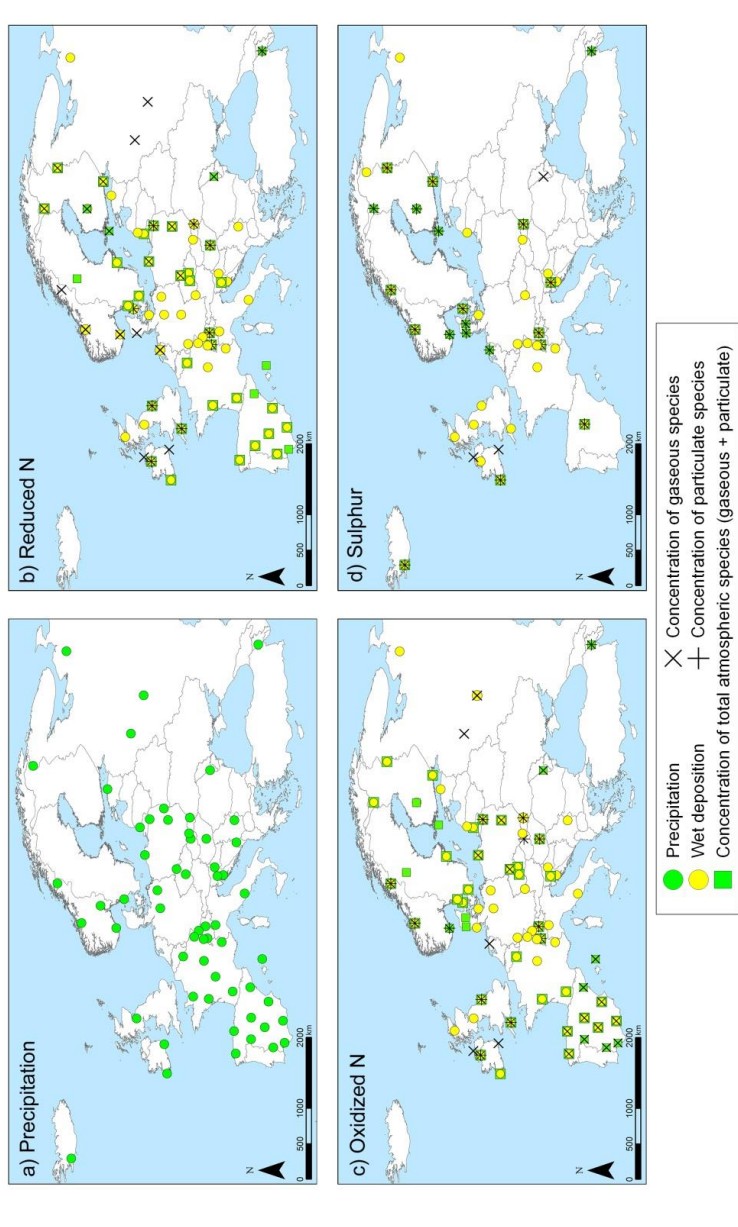


Figure 1:  Monitoring sites with measurements of precipitation (a), reduced N species (b), oxidized N species (c) and S (d)
used in the evaluation of annual modelled values.



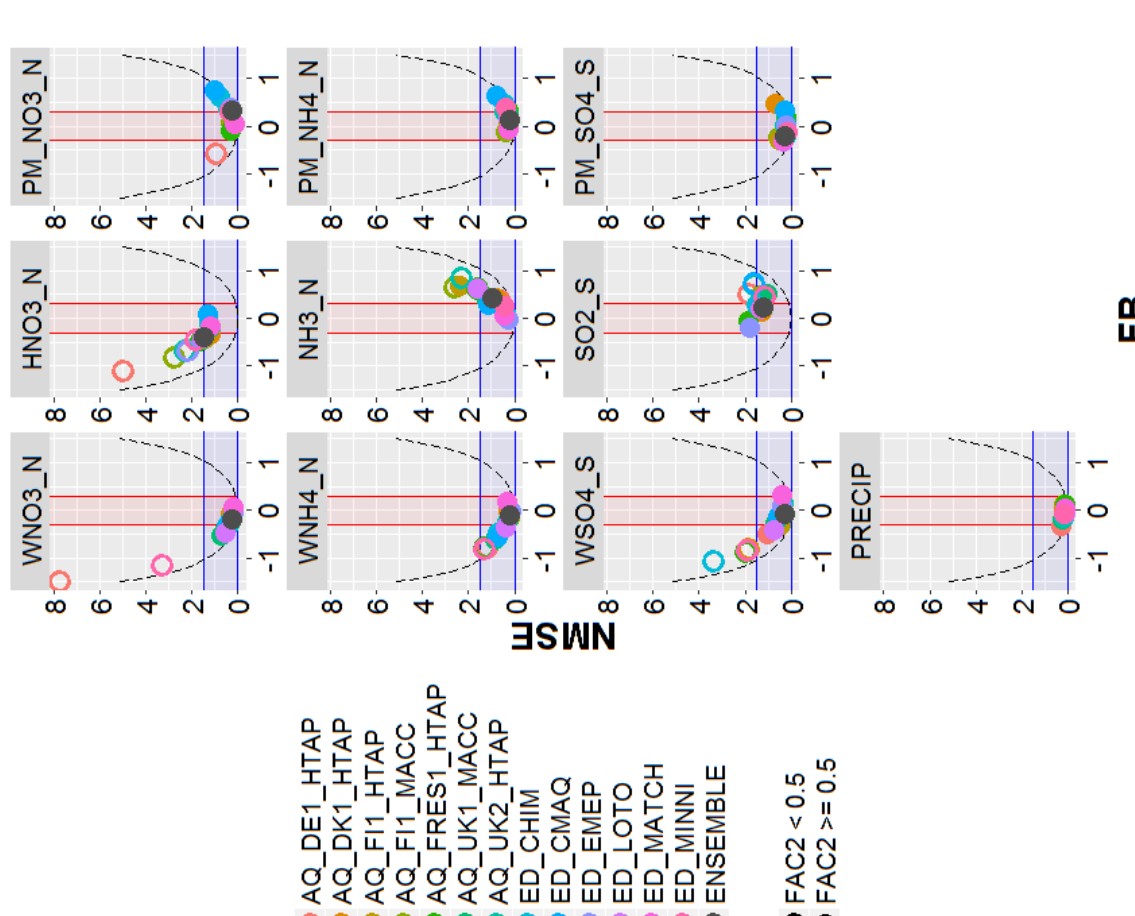


Figure 2:  Statistics (FB, NMSE and FAC2) calculated from annual values of wet deposition, concentration and precipitation
at all available sites. Shaded areas correspond to areas meeting the acceptance criteria of Chang and Hanna (2004) (blue
for NMSE, red for FB). Parabolic dashed lines indicate the theoretical minimum NMSE for a given value of FB. Better
model performance is indicated by points that fall within the blue and red shaded areas and with filled circles.



P

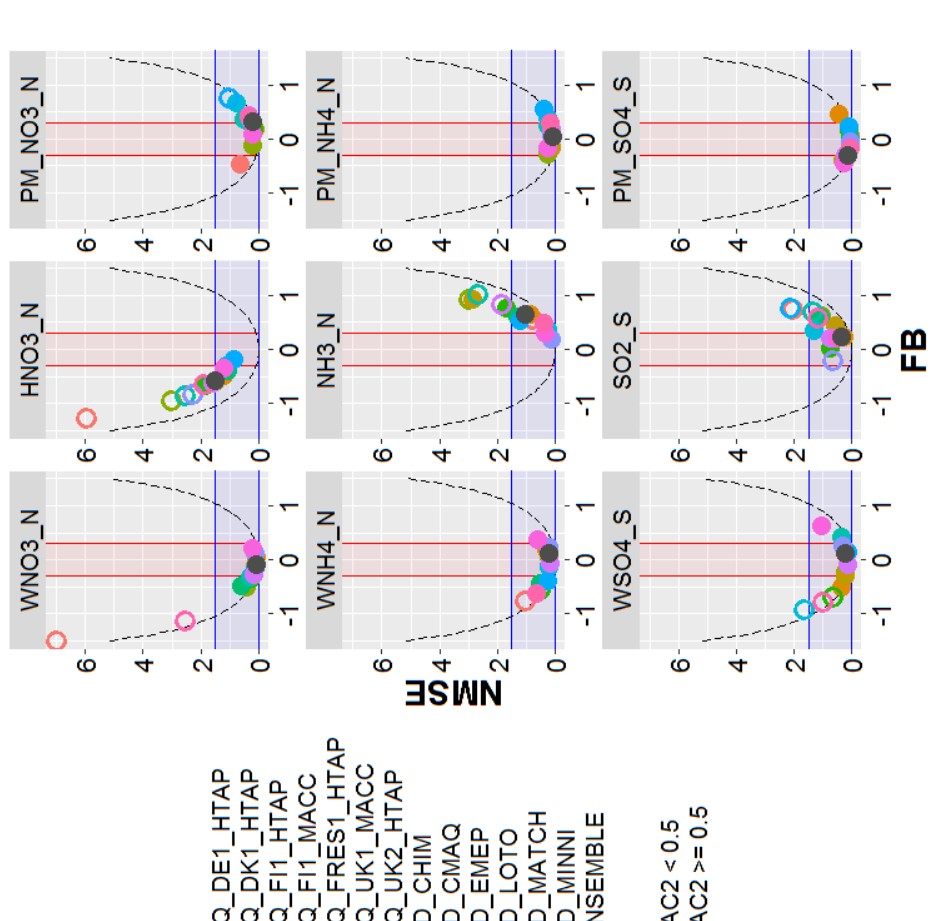

Figure 3: Statistics calculated from annual values (accumulated deposition or average means for air concentration) only at sites with simultaneous measurements of the three related pollutants (e.g. HNO3, PM_NO3 and WNO3) for oxidised N, reduced N and S species. Shaded areas correspond to areas meeting the acceptance criteria of Chang and Hanna (2004) (blue for NMSE, red for FB). Parabolic dashed lines indicate the theoretical minimum NMSE for a given value of FB. Better model performance is indicated by points that fall within the blue and red shaded areas and with filled circles.






P

**Annual deposition of TOTAL N**

Figure 4: Maps of total N (mg N m$^{-2}$) for the models showing acceptable performance for wet N deposition. The ensemble (mean of the models) is shown in right bottom panel



Figure 5: Maps of standard deviation of total N in absolute and relative units (mg N m$^{-2}$; % of annual mean) for the ensemble.



P

Figure 6: Maps of total S (mg N m$^{-2}$) for the models showing acceptable performance for wet S deposition. The ensemble (mean of the models) is included (right bottom map)



P

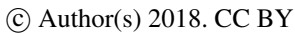


Figure 7: Maps of standard deviation of total S in absolute and relative units (mg S m$^{-2}$; % of annual mean) for the ensemble.





P

Figure 8: Coverage representation of EUNIS level-1 habitat classes within the entire Natura 2000 network versus the buffered areas.






P



Figure 9: Habitat distribution and location of $CL_{exc}$ for the most threatened habitat classes (a: D1 "raised and blanket bogs" and D2 "valley mires, poor fens and transition mires"; b: E4 "alpine and subalpine grasslands"; c: F2 "artic, alpine and subalpine scrub"; d: G3 "coniferous woodlands" and G4 "mixed deciduous and coniferous woodlands"). The surface areas showing a $CL_{exc}$ are represented in red, while the areas with no $CL_{exc}$ are represented ion green.





P

Figure 10:

Figure 10: Proportion of habitat area for which the critical load is exceeded for major terrestrial habitat classes within the Natura 2000 network fpr the base case 2010 (ensemble and AQ_FI1_MACC) and for the EUR, GLO and NAM cases

(AQ_FI1_MACC)



P



Figure 11: Effect on the N deposition in Europe of the reduction of 20% of emissions at global scale (GLO), in Europe (EUR) and in North America (NAM), according to AQ_FI1_MACC (%, left, mgN/m2, right)





Figure 12: Effect on the S deposition in Europe of the reduction of 20% of emissions at global scale (GLO), in Europe (EUR) and in North America (NAM), according to AQ_FI1_MACC (%, left, mgN/m2, right)