# Peer review of "Modelled deposition of nitrogen and sulfur in Europe estimated by 14 air quality model-systems: Evaluation, effects of changes in emissions and implications for habitat protection"

_Atmospheric Chemistry and Physics, 2018_

## Referee Comment (RC1) · Anonymous Referee #1 · 27 Mar 2018

The manuscript is well structured and written. It provides a valuable comparison for modeled deposition of nitrogen and sulfur by fourteen air quality models over Europe. There is a lot of information provided from the evaluation results in the manuscript and the supplementary material. I think the article deserves publication. I have only a few minor comments to be considered by the authors.

In Section 2.1.1 the emissions used are only briefly described. Although there are references provided I would suggest to provide a little more information for Copernicus, HTAP_v2.2 and ECLIPSE_V5 emissions (eg. spatial resolution, temporal resolution).

[Figure]

In Section 2.2 please describe briefly how the statistical measures for each individual station are implemented in smile plots where we see the entire set of stations. It is stated that there is a tendency for the models to underestimate WSO4_S and simultaneously overestimate the gaseous pollutant SO2_S on and annual and monthly basis. Please discuss some possible reasons for this. Is there a possibility for less efficient heterogeneous oxidation of SO2?

In Section 3 it is written that "As can be inferred from AM 2.3, AQ_DK1_HTAP estimate the main contribution from the gas phase,...". To my understanding this holds for AQ_F11_HTAP according to AM2.3 while for AQ_DK1_HTAP the highest contribution comes from the particle phase.

---

## Referee Comment (RC2) · Anonymous Referee #2 · 9 Apr 2018

The authors compare simulated S and N deposition from 14 models. The paper presents extensive information about the performance of the different models and is definitely worth publishing. However, the paper must be improved in several aspects before it can be published. In particular, some more attempts must be made to explain the reasons for the large differences in simulated deposition among some of the models. Furthermore, parts of the paper are not well organized and hard to read.

**Detailed comments:**

[Figure]

Line 100 and Table 2: What is the reason for using such an obsolete version of WRF? Which parameterizations were applied? How does the meteorological input deviate from WRF-Common for those models where a different meteorological input was used and how does this affect the S and N deposition?

Lines 102-110: Information (including tables and figures) about the different boundary conditions and emission data should be given in the supplement. Please summarize quantitative differences in the paper briefly.

Line 135: This section does not describe the model evaluation, just the evaluation method.

Section 2.2: The 'Results and discussion' section includes the evaluation, which should be indicated by a separate subsection. Generally, this section should be better organized by adding subsections.

Lines 231 and 232: 'giving the highest/lowest' sounds somewhat odd.

Line 411: What does 'previously' mean in this context (earlier in this paper, another paper – if so a citation is required)?

Section 6: The 'Conclusions' are just a summary and should at least include some critical comments about the deviations of the simulation results from some of the models and future directions.

Table 2: ED_LOTO: Does the addition of '(nudged)' mean that no nudging was applied for any other model?

Table 3, last line: the order of SO2 and TSO4 should match the order of the nitrogen compounds.

Table 5: The figure caption should be enhanced (add explanations for $CL^*_{exe}$ etc.).

**Figures:**

[Figure]

The order of the figures should be reconsidered. In some places, the discussion would require a different order of the figure. Figures 5 and 7 seem not to be discussed.

**Abbreviations:**

It may increase the readability of the paper if some of the extensively applied abbreviations were replaced by the full text in some places.

Please explain why _N and _S are sometimes added e.g. to TNO3 or WSO4. To me the addtions _N and _S seem to be unnecessary.

Section 3: Why is OND introduced here as a new abbreciation instead of using TNO3 (or TNO3_N)? Same for RN.

Lines 373 – 376: The abbreviations, which are explained here are already used in section 4.1 without explanation.
* * *

---

## Author Response (AR1)

**Reviewers Comments and Author Responses:**

(1) comments from Referees, (2) authors' response, (3) authors' changes in manuscript.

First of all we want to thank the reviewers for their comments and suggestions.

Comments from Referee 1 are referred to as RF1C. Authors' response is indicated by AR:

**Reviewer 1 Comment:**

The manuscript is well structured and written. It provides a valuable comparison for modeled deposition of nitrogen and sulfur by fourteen air quality models over Europe. There is a lot of information provided from the evaluation results in the manuscript and the supplementary material. I think the article deserves publication. I have only a few minor comments to be considered by the authors.

**RFC.1: In Section 2.1.1 the emissions used are only briefly described. Although there are references provided I would suggest to provide a little more information for Copernicus, HTAP_v2.2 and ECLIPSE_V5 emissions (eg. spatial resolution, temporal resolution).**

AR.1: Yes, it's true. We have now added some more information in the text, specifically the spatial and temporal resolution. We have also included this information in Table 2.

**RFC.2: In Section 2.2 please describe briefly how the statistical measures for each individual station are implemented in smile plots where we see the entire set of stations.**

AR.2: Each point in smile plots corresponds to the statistics calculated using the data from all sites combined. We have modified the sentence to clarify this this in lines 152-153:

"For each model simulation and set of sites with observations, the following statistics were calculated (Table 4) for each variable (considering all the values in time and space): "

**RFC.3: it is stated that there is a tendency for the models to underestimate WSO4_S and simultaneously overestimate the gaseous pollutant SO2_S on and annual and monthly basis. Please discuss some possible reasons for this. Is there a possibility for less efficient heterogeneous oxidation of SO2?**

AR.3: Yes, this happens for some models. We included in the text some allusions to an potential underestimation of the aqueous chemistry (559-561):

"The fact that sulfate concentration is also low for several models in Jan and Feb and SO2 somewhat high could be due to an underestimate of the conversion to aerosol (sulfate) via aqueous chemistry, which could be another cause of the excess NH3."

The relation of this to wet deposition would be clear if the efficiency of wet scavenging for SO2 (if overestimated) was lower than that for the sulfates, which in fact is the case for the parameterization used in EMEP model parameterization. But although it's out of the scope of this paper to look into detail the parameterization of all the models, due to the complexity of the variables involved, chemical and meteorological, we have included in the conclusion section the potential occurrence of a low heterogeneous SO2 oxidation efficiency, suggested by the results in this study.

**RFC.4: In Section 3 it is written that "As can be inferred from AM 2.3, AQ_DK1_HTAP estimate the main contribution from the gas phase,. . .". To my understanding this holds for AQ_F11_HTAP according to AM2.3 while for AQ_DK1_HTAP the highest contribution comes from the particle phase.**

AR.4: It's true that this figure in AM 2.3 has not been sufficiently explained, as left (dry deposition from NO2) and middle (dry deposition for HNO3) maps correspond both to gases, and only the one in the right correspond to the particle phase. This could have led to a wrong interpretation, but the statement was correct; for AQ_DK1_HTAP the main contribution to dry deposition comes from the gas phase (in particular from HNO3).This is also valid for AQ_F11_HTAP.We have modified the text slightly to avoid confusion (lines 332-337)

Before:

Significant differences can be found when looking at the gas and particle deposition for the AQMEII3 participants. Two gases, NO2 and HNO3 can contribute to OND. **As can be inferred from AM 2.3, AQ_DK1_HTAP estimate the main contribution from the gas phase, whereas in the case of AQ_TR1_MACC, highest contributions to OND come from the particle phase**. This highlights the importance of making measurements that can shed more light on these processes, providing modelers with data that can be used to parameterize and evaluate the different processes.

Now:

"Significant differences can be found when looking at the gas and particle deposition for the AQMEII3 participants. Two gases, NO2 and HNO3 can contribute to ONDD. **As can be inferred from AM 2.3, in the case of AQ_DK1_HTAP and AQ_F11_HTAP the gas components (NO2 and HNO3) contribute more to ONDD than the particle phase, whereas in the case of AQ_TR1_MACC the largest contributions to ONDD come from the particle phase**. This highlights the importance of making measurements that can shed more light on these processes, providing modelers with data that can be used to parameterize and evaluate the different processes."

**Reviewer 2:**

The authors compare simulated S and N deposition from 14 models. The paper presents extensive information about the performance of the different models and is definitely worth publishing. However, the paper must be improved in several aspects before it can be published. In particular, some more attempts must be made to explain the reasons for the large differences in simulated deposition among some of the models. Furthermore, parts of the paper are not well organized and hard to read.

**RFC.1: Line 100 and Table 2: What is the reason for using such an obsolete version of WRF? Which parameterizations were applied? How does the meteorological input deviate from WRF-Common for those models where a different meteorological input was used and how does this affect the S and N deposition**?

AR.1: The meteorological fields were already available from previous studies in the framework of the EuroCordex climate downscaling programme, where WRF 3.3.1 had been used. Then an optimal setup had been identified and used to re-run the model, applying a grid-nudging towards the ERA-Interim reanalysis above the planetary boundary layer. This WRF simulation was used for the ED project; it was interpolated on the 25 km resolution ED grid and used to drive CHIMERE, EMEP and MINNI.

Due to the variability of parameterizations for the different groups using WRF, (groups are indicated inTable 2), and as they have already been published previously (Solazzo et al., 2017 for AQMWII3 community and Colette et al., 2017 for ED community) we think it is more convenient to include references to these publications, that include the parameterizations used in WRF by each group.

The WRF-common was only used by three models of the ED· project (ED_CHIM, ED_EMEP, ED_MINNI). The other models in ED community used other meteorological drivers. On the other hand, in the AQMEII3 project, meteorological inputs were selected by each modelling group, so there is a wide variability of meteorological information. We focused in this paper on precipitation, since it is a direct driver of wet deposition, by including in the paper statistics for precipitation (annual values in the main text and by month in the AM) for each group, shown as smile plots and tables. We had discussed the performance of models in the original version, saying that they performed well in terms of annual precipitation.

Now we have decided to include a bit more discussion on precipitation, highlighting differences on a temporal basis: including specific ideas such as:

"Smile plots in AM3.5 indicate that some models have larger fractional bias in summer, especially in August, when some models underestimate accumulated precipitation, especially ED_LOTO, AQ_DE1_HTAP, AQ_UK1_MACC, AQ_UK2_HTAP, and the three models using WRF-Common, that is, ED_CHIM, ED_EMEP and ED_MINNI."

**RFC.3: Lines 102-110: Information (including tables and figures) about the different boundary conditions and emission data should be given in the supplement. Please summarize quantitative differences in the paper briefly.**

AR.3: We have included in the text (lines 104-118) and in Table 2 more specific information for emissions and boundary condition (temporal and spatial resolution). Also, we have included a map of differences of emissions of NO2, SO2 and NH3 in the AM 7A) y AM 7B). Later in the paper, we relate differences in models in dry deposition to these maps.

**RFC.4: Line 135: This section does not describe the model evaluation, just the evaluation method.**

AR.4: Yes, as this section was included in 2.1, "Methodology", that's why this part only describes the model evaluation methodology. But as this could result in confusion we have divided section 2; now section 2 is the old 2.1, so methodology is Section 2 and Results is now Section 3. We agree that it is clearer in this way.

**RFC.5: Section 2.2: The 'Results and discussion' section includes the evaluation, which should be indicated by a separate subsection. Generally, this section should be better organized by adding subsections.**

AR.5: We have now divided the manuscript into more Sections/Subsections:

- Section 2: 2 Methodology for the evaluation of wet deposition

Section 3: 3. Results and discussion for wet deposition

- , and we have divided it in 5 subsections:
    o 3.1: Oxidised nitrogen
    o 3.2: Reduced Nitrogen
    o 3.3 Sulfur
    o 3.4 Ensemble
    o 3.5 Joint Discussion

**RFC.6: Lines 231 and 232: 'giving the highest/lowest' sounds somewhat odd.**

AR.6: We have changed this to: "estimating the highest/lowest"

**RFC.7: Line 411: What does 'previously' mean in this context (earlier in this paper, another paper – if so a citation is required)?**

AR.7: Yes, it is a bit confusing. We meant earlier in this paper. We have changed the text to: (Section 5.1)

"As we have previously mentioned, in the framework of AQMEII3 activities and to give scientific support to the HTAP task force, research activities have included an evaluation of the influence of a reduction of emissions in some parts of the Northern Hemisphere on the air quality other regions."

**RFC.8: Section 6: The 'Conclusions' are just a summary and should at least include some critical comments about the deviations of the simulation results from some of the models and future directions.**

AR.8: The conclusions section has now more discussion. We have included some parts that were in the old version in previous sections. We agree that now there are more final comments and some directions to continue investigating in deposition processes of models.

**RFC.9: Table 2: ED_LOTO: Does the addition of '(nudged)' mean that no nudging was applied for any other model?**

AR.9: No, sorry. It's true that this is a bit confusing and unnecessary, as we have not entered in those details for the rest of models. We have removed this "nudged" from the table and we refer to Colette et al. and Solazzo et al. for the WRF specifications.

RFC.10: Table 3, last line: the order of SO2 and TSO4 should match the order of the nitrogen compounds.

AR.10: Yes, we have changed that, thank you.

RFC.11: Table 5: The figure caption should be enhanced (add explanations for CL∗ exe etc.).

AR.11: Done

RFC.12: Figures: The order of the figures should be reconsidered. In some places, the discussion would require a different order of the figure.

AR.12: We have reorganized the paper, by describing first the emission reduction activities and results and after that the effects on vegetation, as graphics on effects included the reduction scenarios. Now we consider that this is much better organized. We moved the figures accordingly.

RFC.13: Figures 5 and 7 seem not to be discussed.

AR.13; Yes, we have now included a reference to them and some discussions (lines 348-356).

RFC.14: Abbreviations: It may increase the readability of the paper if some of the extensively applied abbreviations were replaced by the full text in some places.

AR.14: We have removed some of them from the old Section 3 (now 4).

RFC.15: Please explain why _N and _S are sometimes added e.g. to TNO3 or WSO4. To me the additions _N and _S seem to be unnecessary.

AR.15: We found convenient the use of _N and _S during the treatment of data, due to the diversity of units. To avoid errors in graphics, statistics and therefore in interpretation of results we decided to have very clear variables. We have introduced an explanation to this in Table 1 caption.

RFC.16: Section 3: Why is OND introduced here as a new abbreviation instead of using TNO3 (or TNO3_N)? Same for RN.

AR.16: Well, these were not the same. In this case OND makes reference to dry deposition (D) of oxidized nitrogen, whereas TNO3 is total air concentration of gas and particle. The idea in this old section was to introduce an abbreviation for dry deposition, with a "D". As we see this is still resulting in confussion with have called it now ONDD, that seems to bring more the idea of dry deposition. Same for RND, now changed to RNDD.

RFC.17: Lines 373 – 376: The abbreviations, which are explained here are already used in section 4.1 without explanation.

AR.17: Yes, critical load=CL was not introduced since the first use of this abbreviation. We have included it now in the beginning of old Section 4.

Final comments:
* * *
We have updated the maps with sites, as we noticed some missing sites in the original maps.

**List of major changes:**

1) The manuscript includes now more information on the temporal and spatial resolution of the emissions and boundary conditions. Maps of differences in the emissions used in the ED3 and AQMEII3 projects are included. The discussion on dry deposition makes reference to these differences.
2) The organization for the discussion of wet deposition has changed, with more subsections and a joint discussion section.
3) The order of some parts has been changed; now we show the effects of changes in emissions (reduction scenarios) before the implications for ecosystems.
4) More messages are included in the conclusion section.
5) More discussion on monthly precipitation is included.
6) More discussion on the standard deviation of the ensemble is included.

**Modelled deposition of nitrogen and sulfur in Europe estimated by 14 air quality model-systems: Evaluation, effects of changes in emissions and implications for habitat protection**

Marta G. Vivanco[1,] Mark. R. Theobald[1], Héctor García-Gómez[1], Juan Luis Garrido[1], , Marje Prank[2,3], Wenche Aas[4], Mario Adani[5], Ummugulsum Alyuz[6], Camilla Andersson[7], Roberto Bellasio[8], Bertrand Bessagnet[9], Roberto Bianconi[8], Johannes Bieser[10], Jørgen Brandt[11], Gino Briganti[5], Andrea Cappelletti[5], Gabriele Curci[12], Jesper H. Christensen[11], Augustin Colette[9], Florian Couvidat[9], Kees Cuvelier[13], Massimo D'Isidoro[5], Johannes Flemming[14], Andrea Fraser[15], Camilla Geels[11], Kaj M. Hansen[11], Christian Hogrefe[16], Ulas Im[11], Oriol Jorba[17], Nutthida Kitwiroon[18], Astrid Manders[19], Mihaela Mircea[5], Noelia Otero[20], Maria-Teresa Pay[17], Luca Pozzoli[21], Efisio Solazzo[21], Svetlana Tsyro[22], Alper Unal[23], Peter Wind[22,24] and Stefano Galmarini[21]

[1]Environmental Department, CIEMAT, Madrid, 28040, Spain
[2]Finnish Meteorological Institute, Helsinki, FI00560, Finland
[3]Cornell University, Ithaca, NY, 14850, USA
[4]NILU-Norwegian Institute for Air Research, Kjeller, 2007, Norway
[5]ENEA, Italian National Agency for New Technologies, Energy and Sustainable Economic Development (ENEA), Via Martiri di Monte Sole 4, 40129 Bologna, Italy
[6]Bahcesehir University Engineering and Natural Sciences Faculty. 34353 Besiktas Istanbul, Turkey
[7]SMHI, Swedish Meteorological and Hydrological Institute Norrköping, Norrköping, Sweden
[8]Enviroware srl, Concorezzo, MB, Italy
[9]INERIS, Institut National de l'Environnement Industriel et des Risques, Parc Alata, 60550 Verneuil-en-Halatte, France
[10]Institute of Coastal Research, Chemistry Transport Modelling Group, Helmholtz-Zentrum Geesthacht, Germany
[11]Department of Environmental Science, Aarhus University, Roskilde, DK-4000, Denmark.
[12]Department of Physical and Chemical Sciences, University of L'Aquila, L'Aquila, Italy
[13] Ex European Commission, Joint Research Centre JRC I-21020 Ispra (Va), Italy
[14]European Centre for Medium-Range Weather Forecasts, Reading, UK
[15]Ricardo Energy & Environment, Gemini Building, Fermi Avenue, Harwell, Oxon, OX11 0QR, UK
[16]Computational Exposure Division, National Exposure Research Laboratory, Office of Research and Development, United States Environmental Protection Agency, Research Triangle Park, NC,
[17] BSC, Barcelona Supercomputing Center, Centro Nacional de Supercomputación, Nexus II Building, Jordi Girona, 29, 08034 Barcelona, Spain
[18] Environmental Research Group, Kings' College London, London, UK
[19]Netherlands Organization for Applied Scientific Research (TNO), Utrecht, The Netherlands
[20]IASS, Institute for Advanced Sustainability Studies, Potsdam, Germany
[21] European Commission, Joint Research Centre (JRC, Ispra (VA), Italy
[22]Climate Modelling and Air Pollution Division, Research and Development Department, Norwegian Meteorological Institute (MET Norway), P.O. Box 43, Blindern, N-0313 Oslo, Norway
[23] Eurasia Institute of Earth Sciences, Istanbul Technical University, Turkey
[24]Faculty of Science and Technology, University of Tromsø, Tromsø, Norway

*Correspondence to*: Marta G. Vivanco (m.garcia@ciemat.es)


[revised manuscript text omitted]